# Evaluation of Bone Regenerative Capacity in Rabbit Femoral Defect Using Thermally Disinfected Bone Human Femoral Head Combined with Platelet-Rich Plasma, Recombinant Human Bone Morphogenetic Protein 2, and Zoledronic Acid

**DOI:** 10.3390/biomedicines11061729

**Published:** 2023-06-16

**Authors:** Dina Saginova, Elyarbek Tashmetov, Yevgeniy Kamyshanskiy, Berik Tuleubayev, Denis Rimashevskiy

**Affiliations:** 1Center for Applied Scientific Research, National Scientific Center of Traumatology and Orthopaedics Named after Academician N.D. Batpenov, Astana 010000, Kazakhstan; saginova_d@nscto.kz; 2Department of Surgical Diseases, Karaganda Medical University, Karaganda 100000, Kazakhstan; tuleubaev@qmu.kz; 3Pathology Unit of the University Clinic, Karaganda Medical University, Karaganda 100000, Kazakhstan; kamyshanskiy@qmu.kz; 4Department of Traumatology and Orthopaedics, Peoples’ Friendship University of Russia, Moscow 101000, Russia; denis@rimashevskiy.ru

**Keywords:** bone regeneration, bone graft, platelet-rich plasma, BMP-2, zoledronic acid

## Abstract

This research aimed to assess the effect of bone allograft combined with platelet-rich plasma (PRP), recombinant human bone morphogenetic protein-2 (rhBMP-2), and zoledronic acid (Zol) on bone formation. A total of 96 rabbits were used, and femoral bone defects (5 mm) were created. The rabbits were divided into four groups: (1) bone allograft with PRP (AG + PRP), (2) bone allograft with rhBMP-2 5 μg (AG + BMP-2), (3) bone allograft with Zol 5 μg (AG + Zol), and (4) bone allograft (AG). A histopathological examination was performed to evaluate bone defect healing after 14, 30, and 60 days. The new bone formation and neovascularization inside the bone allograft was significantly greater in the AG + PRP group compared to AG and AG + Zol groups after 14 and 30 days (*p* < 0.001). The use of bone allograft with rhBMP-2 induced higher bone formation compared to AG and AG + Zol groups on days 14 and 30 (*p* < 0.001), but excessive osteoclast activity was observed on day 60. The local co-administration of Zol with a heat-treated allograft inhibits allograft resorption as well as new bone formation at all periods. In conclusion, this study demonstrated that PRP and rhBMP-2, combined with a Marburg bone allograft, can significantly promote bone formation in the early stage of bone defect healing.

## 1. Introduction

The replacement of bone defects is a pressing issue in modern traumatology and orthopedics. Osteogenesis can be stimulated by autografts and bone substitutes, with over 2 million bone graft surgeries being performed worldwide each year [1,2]. Autologous bone is considered the “gold standard” in orthopedics for replacing bone defects caused by various factors. However, the use of autologous bone has its own drawbacks, including filling large bone defects, pain in the donor site, increased operation time, and cosmetic defects [3,4]. This has led to the development of various bone substitute materials, which are structurally similar to bone tissue, as an alternative to autologous bone [5,6,7]. Commercial tissue banks currently offer bone substitute materials, such as cortical cancellous ilium, femoral head, freeze-dried bone substitutes, and decalcified freeze-dried bone, which are sterilized using chemical or physical methods [3,8].

The Marburg Bone Bank prepared bone graft is a type of bone allograft widely used in orthopedic surgery. The Marburg Bone Bank system is based on the thermal disinfection of the femoral bone head and is considered safe, as the bone matrix in the graft provides the necessary osteoconductive properties required for successful bone repair [9,10,11]. However, after being extensively processed, bone allografts lose their innate osteoinductive properties and, as a result, are unable to produce the same clinical outcomes as autologous bone grafts [12,13]. Therefore, there has been a growing interest in using growth factors and morphogens as substances that can provide osteoinductivity to bone substitutes [14,15,16,17].

The addition of bone morphogenetic proteins (BMPs) is an example of such incorporation in bone graft substitutes [18,19]. BMPs are a naturally occurring group of proteins belonging to the transforming growth factor beta (TGF-β) family. They act as cytokines that facilitate the differentiation of mesenchymal cells into bone- and cartilage-forming cells. Among them, recombinant human bone morphogenetic protein-2 (rhBMP-2) is known to be critical in bone formation and healing as it has the ability to induce osteoblast differentiation [18,19,20].

Platelet-rich plasma (PRP) has been used as an autologous blood product in clinical settings to promote tissue regeneration in various types of bone defects and guide bone regeneration with bone grafts [21,22]. PRP is believed to possess the ability to stimulate bone regeneration due to the growth factors released from activated platelets, which have a stimulatory effect on progenitor cells and vascularization at local sites. Platelets may contain unique activators of bone morphogenetic proteins (BMPs) that stimulate the differentiation of progenitor cells into bone-forming cells in laboratory settings [21,23,24].

Bisphosphonates (BP) prevent and treat increased bone resorption in skeletal diseases [25]. Zoledronic acid (Zol) is considered the most potent bisphosphonate in terms of its pharmacological activity and affinity to bone, especially in areas of active bone metabolism [26]. While the effects of Zol on bone resorption have been extensively studied both in vivo and in vitro, its impact on bone formation is still not completely understood and is currently a subject of debate [26,27]. Over the past few decades, some studies have employed specific concentrations of Zol to stimulate osteoregeneration, leading to different findings and varying conclusions [27].

Due to the lack of relevant literature on the combined use of a bone allograft prepared according to the Marburg system and osteoinductive substances, this study aims to evaluate the effect on the bone formation of a bone allograft in combination with platelet-rich plasma, recombined human bone morphogenetic protein-2 (rhBMP-2), and zoledronic acid in rabbit femur defects using histopathological and histomorphometric analyses.

## 2. Materials and Methods

### 2.1. Preparation of Marburg Bone Graft

Three femoral heads were obtained from patients with hip joint osteoarthritis during hip replacement surgery at the Regional Center of Traumatology and Orthopaedics (Karaganda, Kazakhstan). The acquisition of these samples was conducted in accordance with national and ethical regulations, and written informed consent was obtained from all participants involved [28,29].

To prepare the femoral heads for further use, the cartilage was removed using a scalpel and rasp. A drill was utilized in a specialized device to create channels in the bone allograft, following a predetermined template [30]. These drill channels were evenly spaced on the perpendicular walls of the device. Subsequently, the graft underwent a thorough wash in a physiological solution to eliminate any bone debris present within the channels.

Under sterile conditions, the femoral heads were placed in a rigid plastic container designated for disinfection. Approximately 300 mL of sterile Ringer’s solution at room temperature was added to the container, filling it up to a specified mark. To ensure even heat distribution during the disinfection process, a magnetic stirrer was positioned at the bottom of the container, enabling continuous circulation of water. The container was hermetically sealed with a well-fitting lid and transferred to the Lobator SD-2 heating device (Telos GmbH, Marburg, Germany).

For femoral heads with a diameter of 56 ± 1 mm, the manufacturer of the disinfection system guaranteed that a temperature of at least 82.5 °C would be reached at the center of the femoral head for a minimum duration of 15 min [9]. The temperature settings within the device, including the heating phase, plateau, and cooling phase, were pre-programmed by the manufacturer and could not be altered by the user. Throughout the disinfection process, the temperature of the heating device was continuously monitored and recorded at 3 min intervals. Once the disinfection process was completed, a report was generated, containing crucial information such as the device number, identification number, date, and time. This report served as evidence of the completed disinfection process and provided confirmation as to whether the process was carried out in accordance with the required specifications.

Two hours before the experiment, the femoral head was unfrozen (Figure 1) at room temperature and cut into chips. Then, to standardize the mixture of bone allograft with platelet-rich plasma, rhBMP-2, and zoledronic acid, a specific weight was used to ensure a consistent ratio of ingredients: 0.5 g bone allograft/0.5 mL PRP, 0.5 g bone allograft/5 μg rhBMP-2, 0.5 g bone allograft/5 μg Zol.

### 2.2. Preparation of Platelet-Rich Plasma

Prior to each transplantation procedure, approximately 5 mL of blood was collected from the heart and placed in siliconized tubes containing 3.8% sodium citrate at a blood-to-citrate ratio of 9:1 [31]. Blood samples were obtained while the rabbits were under general anesthesia, which was induced by administering an intramuscular injection of Zoletil at a dose of 0.1 mg/kg (Virbac, Fort Worth, TX, USA) and Rometar at a dose of 5 mg/kg (Bioveta- Czech Republic, Ivanovice na Hané, Czech Republic). All animal procedures were performed in accordance with the Guide for the Care and Use of Laboratory Animals [32] and were approved by the University Animal Care Committee (UACC) under protocol No. 27, 27 September 2020. Platelet-rich plasma was obtained through a two-step centrifugation process [24]. The collected blood was initially centrifuged at 900× *g* for 8 min, separating the blood cell component (BCC) in the lower fraction. The BCC fraction was removed, and the remaining material was centrifuged again at 1500× *g* for 5 min to yield platelet-poor plasma (PPP) and PRP. The PRP was obtained by isolating approximately 0.5 mL of the PPP fraction, and subsequently used for impregnation of the bone allograft. A mixture was produced by combining 0.5 mL of liquid PRP with 0.5 g of bone graft.

### 2.3. Preparation of rhBMP-2

A total of 150 μg of recombinant human bone morphogenetic protein-2 (rhBMP-2) (Cusabio, Houston, TX, USA) was mixed with 3 mL of saline solution to form the rhBMP-2 solution (concentration, 50 μg/mL). Subsequently, 100 μL (5 μg) of the aforementioned solution was meticulously introduced into 0.5 g of bone graft utilizing a micropipette, precisely 30 min prior to application [21].

### 2.4. Preparation of Zoledronic Acid

Zoledronic acid (Sun Pharmaceutical Industries Ltd., Chennai, India) with a concentration of 0.05 mg/mL (100 μL) [25] was added to 0.5 g of bone chips by soaking and kept in a sterile container.

### 2.5. Animal Surgery

For this study, 96 adult rabbits weighing 3078 ± 87 g were procured and placed in cages for two weeks to acclimate. During the study, the rabbits were housed at a room temperature of 22 ± 2 °C and maintained at 40–50% humidity under a 12 h light–dark cycle. The rabbits were provided with standard rabbit pellets and tap water throughout the study.

The sample size for the animal experimentation in this study was determined in accordance with Russell and Burch’s bioethical principles of replacement, reduction, and refinement (1959). These principles aim to minimize animal use by using the minimum number of animals necessary to obtain statistically significant results [33].

The rabbits underwent a standardized surgical procedure. Three hours prior to the surgery, the rabbits received an intramuscular (i.m.) injection of gentamycin at a dosage of 0.1 mL/kg (Mapichem, Baar, Switzerland). For anesthesia, the animals were administered an intramuscular injection of Zoletil at a dosage of 0.1 mg/kg (Virbac, Fort Worth, TX, USA) and Rometar at a dosage of 5 mg/kg (Bioveta- Czech Republic, Ivanovice na Hané, Czech Republic). These medications were used to induce and maintain anesthesia throughout the surgical procedure. The hip area was prepared for surgery by shaving and cleaning with an iodine solution. Aseptic techniques were employed, and sterile instruments were used. A 2% lidocaine + epinephrine 1:100,000 solution was diluted to 1% and injected for infiltration. The skin was incised distally, and the muscles were dilated bluntly. A 5 mm drill was used to create bone defects in the metaphysis of the femur to a depth of 10 mm (Figure 2) [34]. Following this procedure, the rabbits were randomly assigned to one of the four experimental groups using simple randomization. The first group received a bone allograft with platelet-rich plasma (PRP) filling in the bone defects (AG + PRP). The second group received a bone allograft with recombinant human bone morphogenetic protein-2 (rhBMP-2) filling in the bone defects (AG + rhBMP-2). In the third group, the bone defects were filled with a bone allograft with zoledronic acid (AG + Zol). Finally, the fourth group (control) received only a bone allograft filling in the bone defects (AG). All surgical procedures were carried out by a trained operator.

The surgical incision was closed using absorbable sutures (4-0 Vicryl, Ethicon, Johnson & Johnson, New Brunswick, NJ, USA). To prevent wound infection following surgery, the rabbits were given intramuscular injections of the antibiotic gentamicin 0.1 mL/kg (MAPICHEM, Switzerland) two times daily for three days postoperatively. Pain relief was provided by administering ketonal 0.04 mL/kg (Sandoz, Ljubljana, Slovenia). The healing process was observed daily after surgery based on a predetermined schedule for several days. At the specified time points of 14, 30, and 60 days, the rabbits were euthanized using a lethal dose of Zoletil 50 mg/mL, and the distal femur was collected for histological analysis (Figure 3). However, on the 14th day, a total of 29 rabbits (AG + PRP—7; AG + rhBMP-2—8; AG + Zol—7; AG—7) were sacrificed because 3 animals were excluded from the experiment. Furthermore, 8 rabbits from each group at each time interval (32 rabbits on the 30 days post surgery and 32 rabbits on the 60th days post surgery) completed the experiment.

### 2.6. Histopathological and Histomorphometric Examination

The bone fragment exhibiting a formed defect underwent histopathological examination subsequent to fixation in 10% neutral buffered formalin for 24 h, followed by decalcification in Biodec R solution for 24 h. The resultant samples were rinsed in phosphate buffer (pH = 7.4) and processed for optimal decalcification. After a bone incision, the tissue was fixed in 10% formalin at 4 °C for 24 h, washed in tap water, dehydrated in graded alcohol concentrations (70%, 90%, 95%, 100%), cleared in xylene, and finally embedded in paraffin blocks. Subsequently, serial longitudinal sections of 5 μm thickness were prepared using a rotary microtome parallel to the sagittal plane and stained using hematoxylin and eosin (for determining the general tissue morphology and cellular composition of the bone defect) and Masson’s trichrome staining (to evaluate the composition of the bone matrix) [35].

-Hematoxylin and eosin staining

Tissue sections were incubated in hematoxylin (Mayer) for 15 min, followed by washing with water for 5 min. Subsequently, the sections were stained with eosin for 1 min.

-Masson’s Trichrome staining

For Masson trichrome staining, a commercially available kit (Trichrome Stain (Masson) Biovitrim TU 9398-001-89079081-2012) was used. After dewaxing and rehydration, the slides were immersed in Bouin’s solution at 56 °C for 15 min. Following this, the slides were washed with tap water for 5 min. Weigert’s hematoxylin was applied for 5 min, followed by another tap water wash for 5 min and a rinse in distilled water. The slides were then stained with Biebrich’s scarlet acid fuchsin for 5 min, washed in distilled water, and incubated in phosphoric-tungstic-phosphomolybdenum acid for 5 min. Aniline blue was applied for 5 min, and finally the slides were fixed in 1% acetic acid for 2 min.

The microscopic evaluation of the preparations was performed using a Zeiss AxioLab 4.0 microscope (Carl Zeiss QEC GmbH, München, Germany) at a magnification of ×400. AxioVision 7.2 software (Carl Zeiss QEC GmbH, München, Germany) was utilized for analyzing and capturing the images. The cellular composition of the bone defect, including osteoclasts, osteoblasts, and osteocytes, was determined by enumerating these cells in each section stained with hematoxylin and eosin. The calculation was based on the number of cells per 1000 cells around the defect zone, and the mean values were expressed to two decimal places for each group. The regeneration of bone defects, measured as a percentage of bone and cartilage tissue, was evaluated within a region delineated by horizontal lines linking the outermost portions of the inner and outer cortical bone layers at the defect boundaries. The proportions of fibrous, cartilage, and bone tissue were quantified morphometrically, expressed as a percentage of the total area of the defect. For every bone defect, we assessed three slices and then calculated the arithmetic mean. The proportion of bone and cartilage tissue in the closure of the defect area was determined by plotting a horizontal line across the outer part of the inner and outer cortical layer of bone at the edges of the defect. Blood vessels were characterized by the presence of erythrocytes in the lumen and endothelial cell lining, and the number of vessels per area of the formed defect was estimated based on 10 fields of view at ×200 magnification. The evaluation was conducted by two certified histologists who were blinded to the group assignment. They assessed various parameters including inflammation, osteogenic cells, tissue composition, angiogenesis, and index of bone defect closure. The index of bone defect closure was calculated as the sum of the relative bone and cartilage area within the bone defect area (Table 1).

### 2.7. Statistical Analysis

The experimental data are presented as the median and interquartile range (Q1–Q3). The Chi-Squared Test with Yates Continuity Correction and Mann–Whitney test were applied for comparing the two groups, while Pearson’s Chi-Squared Test and Kruskal–Wallis Test were used for multiple comparisons. IBM SPSS Statistics 20.0 (IBM Corp., Armonk, NY, USA) and STATISTICA 10 (StatSoft Inc., Tulsa, OK, USA) were used for statistical analysis of the research results. A *p*-value less than 0.05 was considered statistically significant.

## 3. Results

Histological analysis showed no evidence of inflammatory cell infiltration near the allograft in any of the cases. On day 14, the number of osteoblasts in the AG + PRP group was significantly higher than in the AG, AG + rhBMP-2, and AG + Zol groups (*p* < 0.001) (Table 2). The number of osteoblasts in the AG, AG + rhBMP-2, and AG + Zol groups did not differ significantly from each other (*p* > 0.05). On day 30, the number of osteoblasts was significantly higher in the AG + PRP and AG + rhBMP-2 groups than in the AG and AG + Zol groups (Table 2). There was no significant difference in osteoblast numbers between the AG + PRP and AG + rhBMP-2 groups (*p* = 0.819). On day 60, there was no significant difference in osteoblast number among all groups (*p* = 0.730) (Table 2).

On day 14, the number of osteocytes in the AG + PRP group was significantly higher than in the AG, AG + rhBMP-2, and AG + Zol groups (*p* < 0.001) (Table 2). The numbers of osteocytes in the AG + rhBMP-2 and AG + Zol groups did not differ significantly from each other (*p* > 0.05), but both were significantly higher compared to the osteocyte numbers in the AG group (*p* < 0.001). On day 30, the osteocyte numbers in the AG, AG + PRP, and AG + rhBMP-2 groups were significantly higher than in the AG + Zol group (*p* < 0.001). There was no significant difference between AG and AG + rhBMP-2 (*p* = 0.29), although both were significantly lower than in the AG + PRP group (*p* < 0.05) (Table 2). On day 60, osteocyte cell numbers in the AG + PRP, AG + rhBMP-2, and AG groups were significantly higher than in the AG + Zol group (*p* < 0.001) (Table 2). Furthermore, there was no significant difference among the AG, AG + PRP, and AG + rhBMP-2 groups (*p* > 0.05).

On day 14, there was no significant difference in osteoclast numbers among the AG, AG + PRP, AG + rhBMP-2, and AG + Zol groups (*p* > 0.05) (Table 2). On days 30 and 60, the osteoclast numbers were significantly higher in the AG + rhBMP-2 group than in the AG, AG + PRP, and AG + Zol groups (*p* < 0.001) (Table 2). Moreover, there was no significant difference among the AG, AG + PRP, and AG + Zol groups for osteoclast numbers during this period (*p* > 0.05).

At the 14-day interval, new bone formation was observed at the site of the bone defect in three groups: AG + PRP, AG + rhBMP-2, and AG (Figure 4). The AG + PRP group exhibited a significantly greater index of bone defect closure compared to the other groups, as depicted in Figure 5. The newly formed bone adjacent to the graft particles was composed of bone and lacunae containing osteocytes and numerous vascular channels, which were more abundant in the AG + PRP group (Figure 4a). Histologically, the newly formed bone trabecular meshwork was connected to the allograft bone. The bone beams of the newly formed tissue were mostly thin and heterogeneous, with focal bridge-like areas and single contacts, mainly at the poles of the bone beams. In contrast, the AG + Zol group displayed a predominance of fibrous tissue covering the defect area with minimal resorbed allograft fragments (Figure 4c). The fibrous tissue contained single thin-walled vessels and scant infiltrate, with single bone trabeculae found along the edges of the bone plate in a chaotic pattern.

On day 30, the reparative process in the cortical layer of the bone among groups AG + PRP, AG + rhBMP-2, and AG was characterized by a progressive increase in mature bone tissue with minimal fibrosis (Figure 4 and Figure 6). In particular, the AG + PRP and AG + rhBMP-2 groups exhibited a notable prevalence of increased index of bone defect closure and new bone formation compared to the AG and AG + Zol groups (Figure 5b). The bone tissue in the affected area was observed as randomly located bone beams and strands, which formed lamellar structures. Additionally, the bone beams had a high degree of mineralization and demonstrated active longitudinal growth. In terms of newly formed vessels, the AG + PRP group exhibited a higher prevalence compared to the AG + rhBMP-2, AG + Zol, and AG groups (Table 2). At this point in time, the AG + Zol group still demonstrated a predominance of coarse fibrous connective tissue within the defect zone, as was observed on day 14 (Figure 4g and Figure 6g).

At day 60, in groups AG + PRP and AG, complete trabecular bone tissue was observed at the defect site with a normal development of bone trabeculae, which were predominantly composed of spindle-shaped osteocytes (Table 2) (Figure 4 and Figure 6). However, in the AG + rhBMP-2 group, the thickness of the newly formed bone tissue was less than that of the cortical plate outside the defect zone, and a U-shaped depression was formed in the defect area. The newly formed bone tissue was located within the cortical plate and did not spread into the intramedullary space. Furthermore, multinucleated giant cells (osteoclasts) were detected in small lacunae and spaces between allograft fragments and bone beams, which were resorbing the bone tissue (Figure 4j). The intervals between the bone tissues were filled with fibrous tissue without any signs of inflammatory cell infiltration.

In contrast, the defect area in the AG + Zol group was primarily covered with fibrous tissue and bone formation was limited (Table 2). Thin, randomly located, newly formed bone trabeculae extended from the edge of the bone plate into the intramedullary space (Figure 4k and Figure 6k). The boundary between the bone plate and the newly formed bone was evident. The newly formed bone tissue was primarily composed of randomly located, thin bone trabeculae extending into the intramedullary space. The surface of the newly formed bone beams and the edge of the cortical plate defect were surrounded by a fibrous layer without reactive infiltration.

All animals in the study exhibited good tolerance to the surgical procedures. However, it is worth noting that three cases resulted in animals being excluded from the observation during the study period (Figure 3). One rabbit from the AG + Zol group experienced death, while two rabbits (AG + PRP—1; AG—1) suffered from bone fractures. The cause of death for the rabbit could not be determined. However, there were no signs of bone graft rejection, septic complications, or necrotic complications observed in the rabbit. Despite these isolated incidents, the postoperative period progressed without any noticeable complications in 93 animals across all experimental groups. By the end of the study, all animals in the four groups were assessed to be in a satisfactory and healthy condition.

## 4. Discussion

This study examined the use of heat-treated Marburg bone in combination with PRP, rhBMP-2, and zoledronic acid to fill bone defects in an experimental rabbit femur model.

Our research findings reveal that the application of a PRP-perforated allograft combination resulted in superior bone regeneration after 14 days in comparison to the use of rhBMP-2, zoledronic acid, or bone allograft alone to fill the defect. Furthermore, after 30 days, this combination demonstrated better results compared to the usage of zoledronic acid in combination with an allograft and filling the defect with an allograft alone. This fact was confirmed by the analysis of the histological and histomorphometric data of osteogenesis and angiogenesis in the groups. The results obtained are consistent with clinical and experimental studies demonstrating the positive effect of PRP on bone regeneration at an early stage due to the release of several growth factors, such as PDGF, TGF-β, and VEGF [22,23,24]. Furthermore, growth factors contained in PRP stimulate the angiogenesis and proliferation of osteoprogenitor cells only in the very early period after transplantation [36,37,38,39]. In our study, the use of PRP resulted in faster bone formation compared to the other groups in the early stages.

The results we obtained supplement the existing evidence demonstrating that the combination of PRP and thermally treated bone is just as effective as the combination of PRP with other graft materials. Several researchers [40] have suggested that combining PRP does not necessarily result in a significant boost in osteogenesis. This could potentially be due to high-temperature treatment of the biomaterial causing irreversible changes, which may diminish the impact of growth factors present in PRP [41]. Drawing on our findings, we postulate that using PRP with heat-treated bone enhances the initial stage of osteogenesis. Consequently, the early phase of fracture healing with a Marburg bone graft could be expedited through appropriate growth factor stimulation, via the synergistic effect of PRP. This process might lower the occurrence of non-unions and infections.

Concerning the acceleration of bone formation when using a bone allograft with rhBMP-2, it is well known that allografts function as carriers of rhBMP2 [20,42,43]. Previous studies, as well as our results, have shown that the incorporation of rhBMP2 into the allograft significantly accelerates bone formation from an early stage up to 4 weeks after transplantation [44,45]. The bone formation induced by a combined use of a bone allograft with 5 μg of rhBMP2 was higher than that with the allograft alone at day 30. On the contrary, at 60 days, there was a thinning of the cortical plate of the femur at the site of the defect and the presence of many giant cells. This phenomenon is possibly related to the fact that BMPs, in particular rhBMP-2, not only accelerate bone formation but also cause premature bone resorption mediated by osteoclasts through RANKL-RANK signaling, a side effect that is often overlooked [39,46,47]. Further study is underway to investigate if bone resorption and formation occur simultaneously at different dosages and when combined with bisphosphonates. A potential drawback associated with BMP-2 is the formation of bone in non-targeted areas, also known as ectopic bone formation [48,49]. However, throughout the entirety of our observation period in this study, we did not witness any instances of such ectopic bone formation.

The pre-treatment of bone grafts in a bisphosphonate (BP) solution may offer a potential preventive measure against bone graft resorption. The direct influence of BPs on osteoclasts is well established, and this effect on bone formation is noteworthy. The administration of BPs can alter the function of osteoclasts by inducing the secretion of an osteoclastic inhibitory factor by osteoblasts. The regulation of osteoclastic activity is primarily controlled by osteoblastic cells during bone remodeling. BPs enhance the proliferation and differentiation of osteoblasts and reduce apoptotic cell death [25,50,51]. The dosage and application technique of BPs are critical factors that determine the interaction between BPs and bone tissue. Research has revealed that BPs exhibit a biphasic effect on bone cells, with low concentrations stimulating cell proliferation and tissue formation, while high concentrations restrict these processes [26,50,51,52,53,54]. This study used zoledronic acid 0.05 mg/mL in combination with a bone allograft. We found a decrease in resorption, as expected, and a decrease in the amount of new bone in the bone defect at all follow-up periods compared with other groups. In this group, the bone defect was filled mainly with fibrous tissue at all stages (Figure 6). During the remodeling phase, our data support the idea that bisphosphonates may reduce bone resorption and inhibit osteogenic cell activity. Although in previous studies [26,27,53] the authors showed that the use of zoledronic acid 0.05 mg/mL promotes osteogenesis, in this study, the use of zoledronic acid 0.05 mg/mL led to the inhibition of both resorption and bone formation. The application of a thermally treated bone allograft in combination with zoledronic acid has potential applications in joint revision surgery, specifically in scenarios necessitating impaction with bone tissue and prevention of the risk of the endoprosthesis loosening.

The strength of this study is the comparative characterization of the use of a bone allograft prepared according to the Marburg system in combination with PRP, rhBMP-2, and zoledronic acid both in the early stages and in the late stages of bone defect healing. This made it possible to reveal the stimulation of osteoregeneration in the early stages with the use of PRP and rhBMP-2, as well as the inhibition of bone allograft resorption with the use of zoledronic acid. The use of a standardized rabbit femoral defect model and histological and morphometric analysis is also a strength of the study.

We observed notable variability within the data groups, underscoring the importance of developing and standardizing osteoinductive substances. This variation serves as a reminder of the need for such substances that offer controlled and consistent spatial release kinetics of growth factors [55,56], and materials possessing proangiogenic and osteogenic properties [57,58].

The limitations are the absence of an empty defect in the model, which could demonstrate the initial efficiency of the regenerative potential of the bone without any bone grafts, as well as the need to draw blood from the heart before surgery in the group with PRP, which could affect the recovery process after surgical intervention.

Furthermore, the assessment of bone regeneration was solely based on histological analysis. While this provided valuable insights, it is important to acknowledge that other techniques such as micro-CT, immunohistochemistry, and mRNA expression analysis were not employed. These additional methods could have offered a more comprehensive understanding of bone regeneration by examining the structural properties, cellular components, and gene expression patterns of the regenerated bone.

These limitations should be acknowledged when interpreting the findings and should be considered in the design of future studies to address these potential gaps in knowledge.

## 5. Conclusions

In conclusion, the present study showed that PRP and rhBMP-2, in combination with a Marburg bone allograft, can markedly promote bone formation in bone defects at early stages. Although the mechanisms that lead to the stimulation of bone regeneration remain unclear, enhancing the osteoinductivity of bone substitutes may provide a highly relevant approach to clinical bone reconstruction in the near future. In this study, we found a strong effect of bisphosphonates on graft resorption as well as the inhibition of new bone ingrowth into grafts. However, due to the impossibility of direct extrapolation of the results to human conditions, further studies are ongoing in controlled clinical trials.

## Figures and Tables

**Figure 1 biomedicines-11-01729-f001:**
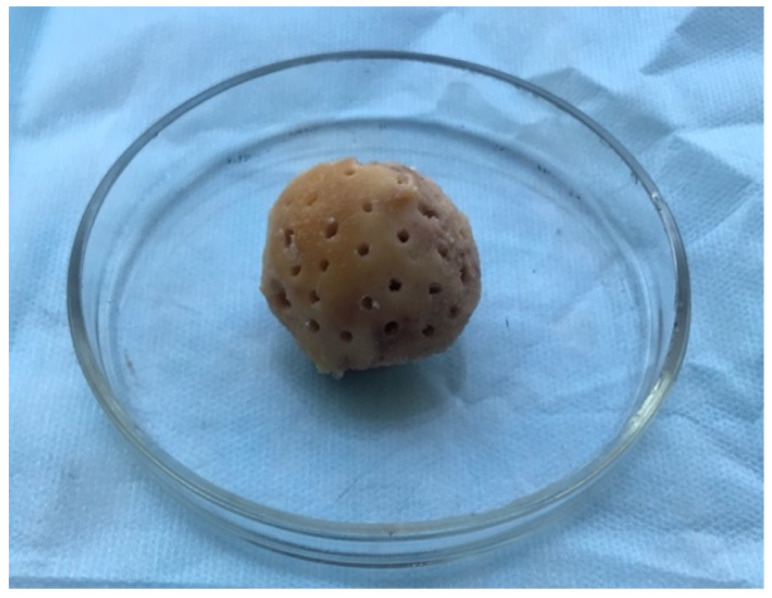
Sterile heat-treated femoral head before application in animal.

**Figure 2 biomedicines-11-01729-f002:**
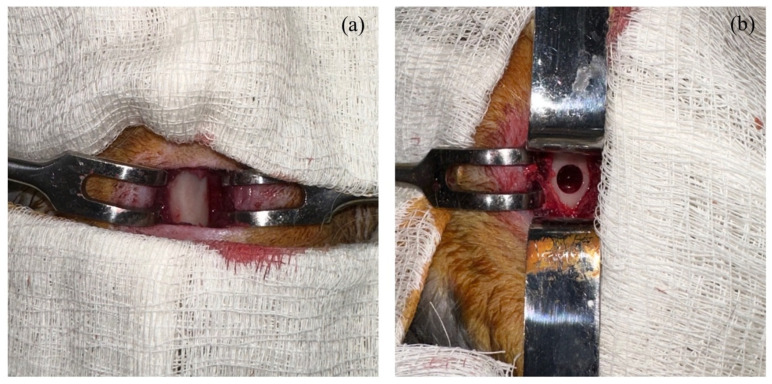
Creation of bone defect in rabbit femur: (**a**) intact bone; (**b**) created bone defect.

**Figure 3 biomedicines-11-01729-f003:**
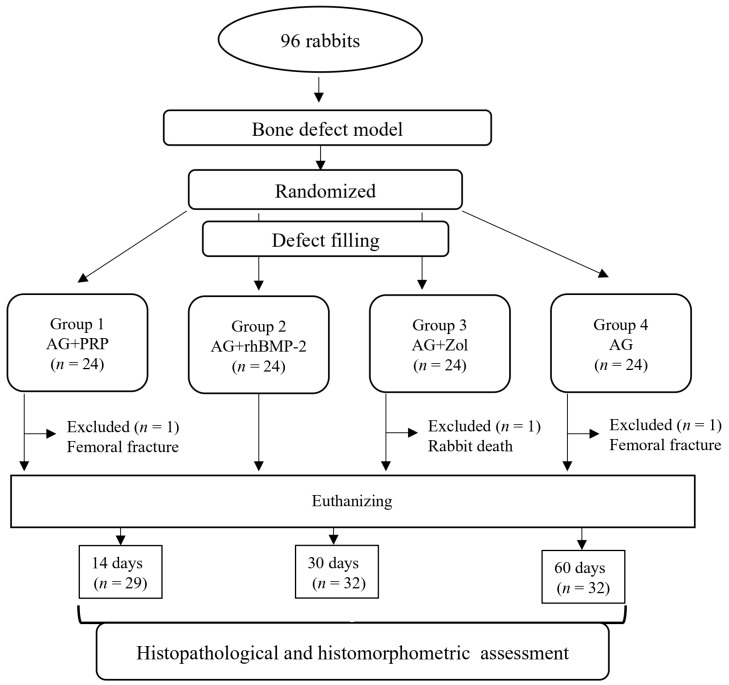
Flow diagram of experiment.

**Figure 4 biomedicines-11-01729-f004:**
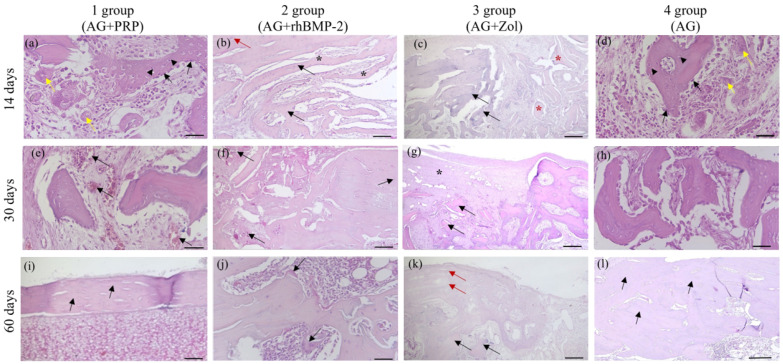
Micrographs of the bone plate defect area from the control and experimental groups at 14, 30, and 60 days after implantation. (**a**,**e**,**i**) AG + PRP group: active reparative microenvironment with active angiogenesis and no inflammation; (**a**) circular strands of osteoblasts (black arrows), mature bone cells (arrowhead), and single osteoclasts (yellow arrows) (HE × 200. Scale bar, 100 µm); (**e**) active angiogenesis (black arrows) (HE × 200. Scale bar, 100 µm); (**i**) complete closure of defect (black arrow) and bone marrow remodeling with histopattern normal cell composition (HE × 40. Scale bar, 500 µm). (**b**,**f**,**j**) AG + rhBMP-2 group: moderate reparative microenvironment without active inflammation with persistence of giant multinucleated osteoclasts in microlacunae of newly formed bone tissue; (**b**) newly formed bone tissue (asterisk) with mature bone cells (black arrows) and osteoclasts (red arrow) (HE × 100. Scale bar, 200 µm); (**f**) weak angiogenesis (black arrows), edge of bone defect (HE × 100. Scale bar, 200 µm); (**j**) osteoclasts (black arrows) (HE × 100. Scale bar, 200 µm). (**c**,**g**,**k**) AG + Zol group: weak reparative microenvironment with weak allograft remodeling; (**c**) single chaotically located bone trabeculae (black arrows) and fragments of lysed allograft (asterisks) are defined (HE × 40. Scale bar, 500 µm); (**g**) fragments of allograft (black arrows) with weak microenvironment (asterisk) (HE × 40. Scale bar, 500 µm); (**k**) newly formed bone tissue from bone defect edge (red arrow) with integrated remodeled allograft (black arrows) (HE × 100. Scale bar, 200 µm). (**d**,**h**,**l**) AG group: moderate reparative microenvironment without active inflammation; (**d**) circular bands of osteoblasts (black arrows), mature bone cells (arrowheads), and single osteoclasts (yellow arrow) (HE × 200. Scale bar, 100 µm); (**h**) newly formed bone beams (HE × 200. Scale bar, 100 µm); (**l**) complete closure of the defect (black arrow) (HE × 200. Scale bar, 100 µm).

**Figure 5 biomedicines-11-01729-f005:**
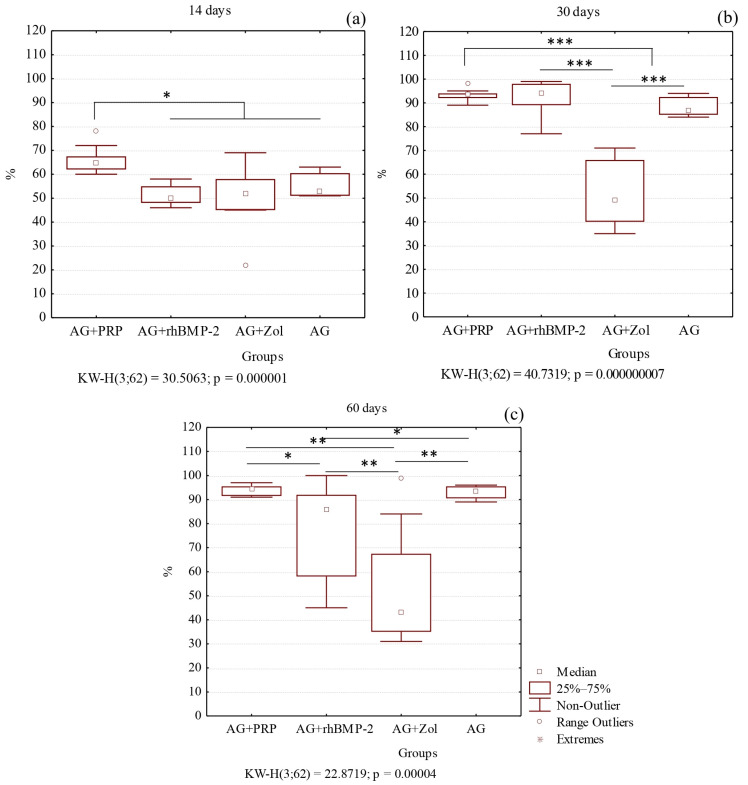
Index of bone defect closure (%) in femoral defects: (**a**) 14 days; (**b**) 30 days; (**c**) 60 days. AG + PRP, bone allograft with platelet-rich plasma; AG + rhBMP-2, bone allograft with recombinant human bone morphogenetic protein-2; AG + Zol, bone allograft with zoledronic acid; AG, bone allograft. (* *p* < 0.05; ** *p* < 0.01; *** *p* < 0.001).

**Figure 6 biomedicines-11-01729-f006:**
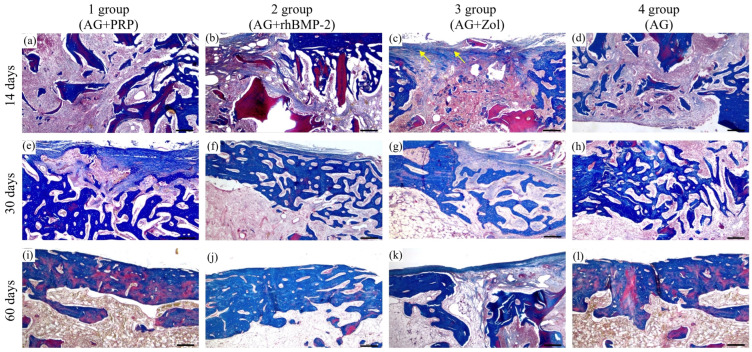
Micrographs of stage-specific closure of the bone plate defect in the control and experimental groups at 14, 30, and 60 days after implantation. (**a**,**e**,**i**) AG + PRP group: active osteogenesis with formation of bands of newly formed bone tissue on the 14th day and complete closure of the defect by mature bone tissue on the 30–60th day (Masson’s trichrome × 40. Scale bar, 500 µm). (**b**,**f**,**j**) AG + rhBMP-2 group: active osteogenesis with uneven closure of the defect thickness by mature bone tissue on day 30–60 (Masson’s trichrome × 40. Scale bar, 500 µm). (**c**,**g**,**k**) AG + Zol group: heterogeneity of the osteogenesis pattern (weak to moderate) with weak allograft remodeling on day 14 and partial closure of the bone plate defect by the newly formed bone tissue and fibrous tissue (yellow arrow) on day 30–60 (Masson’s trichrome × 40. Scale bar, 500 µm). (**d**,**h**,**l**) AG group: moderate osteogenesis with the formation of bands of newly formed bone tissue on the 14th day and complete closure of the defect with mature bone tissue on the 60th day (Masson’s trichrome × 40. Scale bar, 500 µm).

**Table 1 biomedicines-11-01729-t001:** Histopathological parameters of bone defect healing score.

Histological Score	
Inflammation	Polymorphonuclear leukocytes *
	Lymphocytes *
	Macrophages/Histiocytes *
Osteogenic cells	Osteoblasts **
	Osteocytes **
	Osteoclasts **
Index of bone defect closure ***	Bone defect closure area (%)
Tissue composition	Fibrous tissue (%)
	Cartilage (%)
	Bone (%)
Angiogenesis	Neovascularization ****

*—assessment of cellular infiltrate was carried out on 100 cells by summing the average values of different cell types in the area of the defect zone. **—assessment of the cellular composition was carried out for 1000 cells by summing the average values of different types of cells in the area of the defect zone. ***—index of bone defect closure was calculated by summing up the % relative area of bone and cartilage tissue in % of the total area of the defect zone. ****—assessment of the number of newly formed vessels was carried out on the area of the formed defect calculated for 10 fields of view.

**Table 2 biomedicines-11-01729-t002:** (A) Histopathological evaluation of bone defect healing after 14 days. (B) Histopathological evaluation of bone defect healing after 30 days. (C) Histopathological evaluation of bone defect healing after 60 days.

(A)				
	1 Group (AG + PRP)	2 Group (AG + rhBMP-2)	3 Group (AG + Zol)	4 Group (AG)
14 days				
Osteoblasts	412.0 (410.0; 465.8) p1 = 0.0001 p2 = 0.0001 p3 = 0.0001 p4 = 0.0001	273.0 (267.0; 304.0) p5 = 0.5059 p6 = 0.1975	268.0 (262.8; 289.0) p7 = 0.1134	298.0 (281.5; 310.5)
Osteocytes	335.0 (317.8; 366.8) p1 = 0.0001 p2 = 0.0001 p3 = 0.0001 p4 = 0.0001	250.0 (241.3; 251.8) p5 = 0.6929 p6 = 0.0001	265.5 (240.8; 287.0) p7 = 0.0001	217.0 (193.0; 229.5)
Osteoclasts	8.0 (5.0; 11.0) p1 = 0.4196 p2 = 0.1767 p3 = 0.2582 p4 = 0.5591	10.5 (7.3; 13.5) p5 = 0.5605 p6 = 0.2444	11.0 (5.8; 11.3) p7 = 0.5217	8.5 (4.5; 11.0)
Fibrous tissue	35.0 (32.8; 38.0) p1 = 0.0001 p2 = 0.0001 p3 = 0.0001 p4 = 0.0001	50.0 (45.8; 51.8) p5 = 0.8192 p6 = 0.0235	48.5 (42.8; 55.0) p7 = 0.1134	47.0 (40.8; 49.0)
Cartilage tissue	5.5 (3.8; 9.0) p1 = 0.0001 p2 = 0.0001 p3 = 0.0001 p4 = 0.0001	12.5 (12.0; 14.5) p5 = 0.0001 p6 = 0.0078	16.0 (16.0; 18.8) p7 = 0.0001	11.5 (10.8; 12.3)
Bone tissue	58.0 (54.5; 62.8) p1 = 0.0001 p2 = 0.0003 p3 = 0.0001 p4 = 0.0001	38.0 (36.3; 40.0) p5 = 0.1400 p6 = 0.0178	36.0 (27.5; 38.0) p7 = 0.0009	41.5 (38.8; 48.5)
Vessels	39.0 (33.8; 41.3) p1 = 0.0001 p2 = 0.0001 p3 = 0.0001 p4 = 0.0001	20.0 (18.0; 27.3) p5 = 0.0001 p6 = 0.0306	5.0 (5.0; 10.3) p7 = 0.0001	16.0 (14.3; 20.0)
**(B)**				
	**1 Group** **(AG + PRP)**	**2 Group** **(AG + rhBMP-2)**	**3 Group** **(AG + Zol)**	**4 Group** **(AG)**
**30 days**				
Osteoblasts	410.5 (401.5; 446.8) p1 = 0.0001 p2 = 0.8192 p3 = 0.0001 p4 = 0.0129	417.0 (337.3; 457.5) p5 = 0.0047 p6 = 0.2124	276.5 (233.3; 354.0) p7 = 0.0104	364.5 (316.5; 398.5)
Osteocytes	438.5 (428.5; 455.3) p1 = 0.0001 p2 = 0.0014 p3 = 0.0001 p4 = 0.0001	425.5 (396.3; 427.8) p5 = 0.0001 p6 = 0.2987	255.0 (230.0; 371.0) p7 = 0.0001	408.5 (389.5; 421.5)
Osteoclasts	6.5 (5.8; 7.0) p1 = 0.0001 p2 = 0.0001 p3 = 0.2278 p4 = 0.8653	22.5 (17.5; 24.0) p5 = 0.0001 p6 = 0.0001	11.0 (2.0; 11.0) p7 = 0.5465	6.5 (2.8; 11.0)
Fibrous tissue	6.5 (6.0; 8.0) p1 = 0.0001 p2 = 0.9008 p3 = 0.0001 p4 = 0.0018	6.0 (2.0; 11.0) p5 = 0.0001 p6 = 0.0306	51.0 (35.0; 60.0) p7 = 0.0001	12.0 (8.5; 14.5)
Cartilage tissue	8.5 (6.8; 9.3) p1 = 0.0001 p2 = 0.0026 p3 = 0.0007 p4 = 0.7063	5.5 (1.0; 7.8) p5 = 0.0001 p6 = 0.0199	12.0 (11.0; 13.5) p7 = 0.1134	9.0 (5.5; 13.0)
Bone tissue	84.5 (83.8; 85.3) p1 = 0.0001 p2 = 0.1559 p3 = 0.0001 p4 = 0.0006	89.5 (85.0; 93.3) p5 = 0.0001 p6 = 0.0018	34.0 (29.0; 53.8) p7 = 0.0001	80.5 (76.0; 82.5)
Vessels	31.0 (25.0; 41.3) p1 = 0.0001 p2 = 0.0003 p3 = 0.0000 p4 = 0.0041	19.0 (14.8; 21.8) p5 = 0.0019 p6 = 0.0673	10.0 (8.0; 11.0) p7 = 0.0001	20.5 (19.0; 23.3)
**(C)**				
	**1 Group** **(AG + PRP)**	**2 Group** **(AG + BMP-2)**	**3 Group** **(AG + Zol)**	**4 Group** **(AG)**
**60 days**				
Osteoblasts	405.0 (350.8; 421.3) p1 = 0.7302 p2 = 0.8191 p3 = 0.6511 p4 = 0.4286	383.0 (341.0; 427.3) p5 = 0.3286 p6 = 0.3714	396.0 (370.0; 424.0) p7 = 0.9989	402.5 (357.0; 441.3)
Osteocytes	422.0 (411.0; 435.0) p1 = 0.0001 p2 = 0.8679 p3 = 0.0001 p4 = 0.6511	412.5 (412.0; 453.0) p5 = 0.0001 p6 = 0.6511	301.0 (212.0; 314.3) p7 = 0.0001	422.5 (413.8; 431.5)
Osteoclasts	5.5 (4.8; 7.0) p1 = 0.0001 p2 = 0.0001 p3 = 0.7063 p4 = 0.9399	19.0 (14.3; 21.8) p5 = 0.0001 p6 = 0.0001	6.0 (3.5; 8.3) p7 = 0.7630	6.0 (3.8; 7.0)
Fibrous tissue	5.5 (4.8; 8.3) p1 = 0.0001 p2 = 0.0142 p3 = 0.0000 p4 = 0.5465	14.0 (8.8; 41.0) p5 = 0.0004 p6 = 0.0258	57.0 (47.0; 64.0) p7 = 0.0001	6.5 (4.8; 8.8)
Cartilage tissue	11.5 (10.0; 13.3) p1 = 0.3017 p2 = 0.0964 p3 = 0.8802 p4 = 0.4509	5.0 (2.5; 10.0) p5 = 0.2279 p6 = 0.1345	11.5 (9.5; 12.5) p7 = 0.4509	10.5 (9.8; 11.8)
Bone tissue	81.5 (80.8; 84.3) p1 = 0.0001 p2 = 0.7711 p3 = 0.0000 p4 = 0.8081	81.0 (49.3; 89.3) p5 = 0.0009 p6 = 0.9337	32.5 (26.8; 49.0) p7 = 0.0001	81.0 (80.8; 82.3)
Vessels	14.5 (10.3; 15.8) p1 = 0.4180 p2 = 0.6475 p3 = 0.4660 p4 = 0.5316	15.5 (9.0; 17.0) p5 = 0.2617 p6 = 0.6776	12.5 (8.0; 15.0) p7 = 0.7063	13.0 (11.0; 15.8)

Note: p is the significance level; p1 < 0.05—statistically significant difference between all groups; p2 < 0.05—statistically significant difference compared to AG + PRP and AG + rhBMP-2; p3 < 0.05—statistically significant difference compared to baseline AG + PRP and AG + Zol; p4 < 0.05—statistically significant difference compared to the corresponding values in control samples AG + PRP and AG; p5 < 0.05—statistically significant difference compared to the corresponding values in control samples AG + rhBMP-2 and AG + Zol; p6 < 0.05—statistically significant difference compared to baseline AG + rhBMP-2 and AG; p7 < 0.05—statistically significant difference compared to the corresponding values in control samples AG + Zol and AG.

## Data Availability

The datasets generated and/or analyzed during the current study are available from the corresponding author upon reasonable request.

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
