# Peer review of "Evaluation of Bone Regenerative Capacity in Rabbit Femoral Defect Using Thermally Disinfected Bone Human Femoral Head Combined with Platelet-Rich Plasma, Recombinant Human Bone Morphogenetic Protein 2, and Zoledronic Acid"

_biomedicines, 2023, doi:10.3390/biomedicines11061729_

Round 1

Reviewer 1 Report

The authors well planned, conducted and described experiments on osteoregeneration in the field of implantation of the studied materials on animals. However, there is very little information on the physicochemical properties of the implants and coatings used. For a better understanding, it would be nice to provide microphotographs of the topography of implant surfaces, obtained, for example, using scanning electron microscopy or other methods, for example as in https://doi.org/10.1186/s40729-022-00427-1 or any other. Or authors should refer to previous research in this area. In addition, it would be useful for a better understanding of the overall picture of the experiment to know how stable the implant modification is in vivo.

Author Response

Q: The authors well planned, conducted and described experiments on osteoregeneration in the field of implantation of the studied materials on animals. However, there is very little information on the physicochemical properties of the implants and coatings used. For a better understanding, it would be nice to provide microphotographs of the topography of implant surfaces, obtained, for example, using scanning electron microscopy or other methods, for example as in https://doi.org/10.1186/s40729-022-00427-1 or any other. Or authors should refer to previous research in this area. In addition, it would be useful for a better understanding of the overall picture of the experiment to know how stable the implant modification is in vivo.

Answer:  We want to express our sincere gratitude for taking the time to review our manuscript. Your thoughtful comments and suggestions are greatly appreciated and have given us the opportunity to improve the quality of our work. 

We understand your suggestion for the use of electron microscopy in the study. While this indeed could have added another dimension to our research, we made a conscious decision not to include electron microscopy in this particular study. Our primary focus was on exploring the histopathological aspects of bone defect regeneration through a specific lens – using bone allograft harvested according to the Marburg bone bank system in combination with osteoinductive substances.

Regarding the detailed description of the implant surface and the physicochemical features of the bone allograft, these have been extensively characterized and reported in prior literature by other authors (https://doi.org/10.1007/s10561-020-09832-5; doi: 10.1093/rb/rby002), hence we did not emphasize them in this study. We deemed it essential to concentrate on our specific research objectives and to contribute novel insights in this particular field.

However, we completely understand your point of view and we agree that it would be beneficial for the readers to have some context. Therefore, we have decided to add references to the previous works where these aspects have been detailed, enabling readers to delve deeper into these topics if they wish.

Thank you once again for your valuable feedback. Your contribution has helped us to improve our work and present it in a way that is more comprehensive and beneficial for our readers. We look forward to hearing any other comments or suggestions you might have, and we hope that you will find the revised manuscript acceptable for publication.

Reviewer 2 Report

In this manuscript, Saginova and her colleagues report on their evaluation of the bone formation effect using bone allograft in combination with platelet-rich plasma, recombinant human bone morphogenetic protein-2 (rhBMP-2), or zoledronic acid in rabbit femur defects. However, I find the novelty of this work unclear. The authors mention the lack of relevant literature on the combined use of a bone allograft prepared according to the Marburg system and osteoinductive substances as a reason for their study's novelty. However, this statement is misleading and does not explain the novelty sufficiently. Additionally, the manuscript lacks a clear explanation of the novelty and hypothesis, making the significance of the work unclear to me.

Below are some detailed comments:

1.      Combinations of allograft with PRP, rhBMP-2, or Zol have been reported in the literature. For example, PRP was studied in the Journal of Orthopaedic Research 2004, 22, 653-658, rhBMP-2 in The Journal of Bone & Joint Surgery 88(7), 1431-1441, and Zol in BMC Musculoskeletal Disorders 2012, 13, 240. What is the significance of studying this again? What additional information do the authors want to bring in?

2.      The thermal disinfection method appears to be a highlight of this work; however, this method was applied to all the experimental groups, and there is no experiment using the conventional disinfection method as a control. Therefore, there are no results supporting the claim that the thermal method is superior to conventional methods.

3.      The manuscript only includes histological analysis of bone formation. I suggest that the authors conduct additional analyses, such as Micro-CT, to evaluate the outcomes of bone formation.

4.      The manuscript does not provide details on how the bone defect healing rate shown in Figure 5 was calculated. Moreover, the results in this figure are difficult to interpret due to the wide error bars (e.g., Fig. 5c), indicating high variation within each group. Consequently, drawing reliable conclusions from these results would be challenging.

5.      Details characterization of the heat-treated femoral head allograft will help to increase the quality of this work.

In conclusion, this manuscript does not present novel work of high quality and does not meet the standard for publication. Therefore, I recommend rejecting this work.

moderate

Author Response

Q1:      Combinations of allograft with PRP, rhBMP-2, or Zol have been reported in the literature. For example, PRP was studied in the Journal of Orthopaedic Research 2004, 22, 653-658, rhBMP-2 in The Journal of Bone & Joint Surgery 88(7), 1431-1441, and Zol in BMC Musculoskeletal Disorders 2012, 13, 240. What is the significance of studying this again? What additional information do the authors want to bring in?

Answer: Thank you for your constructive feedback and the insightful questions you raised about our manuscript. Your comments have given us the opportunity to improve our study and provide better clarity to our readers. All changes in the text are highlighted in yellow. Please find below our detailed responses to your queries:

We appreciate your attention to the methodological aspects of our study. As you correctly noted, previous research has used fresh frozen, lyophilized, decalcified, and other types of bone grafts as scaffolds. However, these studies cannot be directly correlated to our investigation, as the processing method significantly affects the biological, mechanical qualities, and architectural integrity of the bone graft.

Our study specifically addresses the application of bone allografts prepared according to the Marburg system with PRP, rhBMP-2, and zolendronic acid. No such specific study was found in the existing literature, which motivates our research. Our study contributes to the understanding of bone defect regeneration dynamics using the aforementioned combination, providing new insights into reparative processes using different bone substitutes.

Regarding the publications you mentioned, we acknowledge their relevance but also stress that they differ from ours in critical ways. Here, Weaddress each one:

  • Jensen TB, Rahbek O, Overgaard S, Søballe K. Platelet rich plasma and fresh frozen bone allograft as enhancement of implant fixation. An experimental study in dogs. J Orthop Res. 2004 May;22(3):653-8. doi: 10.1016/j.orthres.2003.10.006.

The design of this study involved modeling a bone defect and filling it with a fresh frozen bone allograft (from another dog) in combination with auto PRP. In the study, 1-step centrifugation at 4000 rpm for 20 min was used for PRP harvesting. In our study, the bone defect was filled with a thermally treated human femoral head in combination with auto PRP. In our study, 2-step centrifugation was used, which is more effective compared to 1-step centrifugation.

  • Jones AL, Bucholz RW, Bosse MJ, Mirza SK, Lyon TR, Webb LX, Pollak AN, Golden JD, Valentin-Opran A; BMP-2 Evaluation in Surgery for Tibial Trauma-Allgraft (BESTT-ALL) Study Group. Recombinant human BMP-2 and allograft compared with autogenous bone graft for reconstruction of diaphyseal tibial fractures with cortical defects. A randomized, controlled trial. J Bone Joint Surg Am. 2006 Jul;88(7):1431-41. doi: 10.2106/JBJS.E.00381.

The second publication examined a bone defect filled with fresh-frozen bone allograft combined with rhBMP-2. Our study, however, focused on a bone defect filled with a heat-treated bone graft in combination with rhBMP-2.

  • Belfrage O, Isaksson H, Tägil M. Local treatment of a bone graft by soaking in zoledronic acid inhibits bone resorption and bone formation. A bone chamber study in rats. BMC Musculoskelet Disord. 2012 Dec 5;13:240. doi: 10.1186/1471-2474-13-240.

In this study a freshly frozen bone allograft with zoledronic acid was used. The study design involved soaking the bone allograft in 3 ml of 0.5 mg/ml zolendronic acid solution.  In our study, the bone defect was filled with a thermally treated human femoral head in combination with zolendronic acid. The bone graft was soaked in 100 μL of 0.05 mg/ml zolendronic acid (5μg per 0.5 g bone). Although zoledronic acid was used in both studies, the concentration of zoledronic acid, the defect model, and the defect filler were different.

All 3 studies utilized fresh-frozen bone allograft. Fresh and frozen allograft materials possess superior osteoinductive properties but are rarely used nowadays due to the higher risk of a host immunogenic response, limited shelf life, and increased risk of disease transmission. Moreover, the freezing process also delays the vascularization of allografts after implantation, which might be detrimental, as vascularity has been identified as a central component influencing bone healing and necessary to effective graft repair.  

Marburg heat-treated bone allografts undergo a specific heat treatment process that aims to reduce the risk of disease transmission and immunogenicity. This process generally preserves the biomechanical properties of the graft.

These differences in design and methods are vital, and thus, their results are not directly transferable to our study.

Q2:      The thermal disinfection method appears to be a highlight of this work; however, this method was applied to all the experimental groups, and there is no experiment using the conventional disinfection method as a control. Therefore, there are no results supporting the claim that the thermal method is superior to conventional methods.

Answer: Our study focused on comparing the use of bone allografts prepared according to the Marburg system with PRP, rhBMP-2, and zolendronic acid to pure bone grafts. We did not specifically compare thermally treated allografts with other types of treatment in this study. That said, other studies comparing thermally treated allografts with other treatments do exist in the literature (doi: 10.1093/rb/rby002; DOI: 10.1159/000335647; DOI: 10.1007/s10561-014-9442-0; DOI: 10.1302/0301-620X.89B5.19039) and we could consider them in future research.

Q3:      The manuscript only includes histological analysis of bone formation. I suggest that the authors conduct additional analyses, such as Micro-CT, to evaluate the outcomes of bone formation.

Answer: Thank you for your comment. In our research, we centered our attention on a microscopic examination of the histological and histomorphometric characteristics of the cellular components within the osteogenic microenvironment during bone regeneration. This was conducted using a thermally treated allograft in conjunction with several bioactive factors, including PRP, rhBMP-2, and zoledronic acid. We acknowledge the potential benefits of incorporating noninvasive imaging techniques to supplement our microscopic observations, as they could offer an expanded perspective on the process of bone regeneration. Incorporation of such imaging techniques is a suggestion worth considering for forthcoming studies.

Q4:      The manuscript does not provide details on how the bone defect healing rate shown in Figure 5 was calculated. Moreover, the results in this figure are difficult to interpret due to the wide error bars (e.g., Fig. 5c), indicating high variation within each group. Consequently, drawing reliable conclusions from these results would be challenging.

Answer: Thank you for your comment. It has been corrected in the manuscript.

We acknowledge that representing data as a range, which heavily depends on extreme values, can make visual interpretation challenging and assessing variability within a group difficult. To counteract this and provide a more transparent representation, we opted to present our data as an interquartile range. This measure, less affected by extreme values, considers the asymmetric distribution and the presence of outliers in the original sample, thus providing a more robust picture of variability.

The regeneration of bone defects (expressed as a percentage of bone and cartilage tissue) was evaluated within an area defined by horizontal lines connecting the outermost parts of the inner and outer cortical bone layers at the defect margins. The morphometric quantification of fibrous, cartilage, and bone tissue was carried out as a percentage of the total defect area. For each defect, three slices were assessed and the arithmetic mean was then calculated.

Q5:      Details characterization of the heat-treated femoral head allograft will help to increase the quality of this work.

Answer: Our article provided a standard algorithm for femoral head blanks according to the Marburg system. For a more comprehensive understanding of this technology, we recommend the references listed in our article or directly reaching out to the corresponding author.

We hope our responses adequately address your queries and look forward to your continued interest in our work.

Reviewer 3 Report

‘Evaluation of bone regenerative capacity in rabbit femoral defect using thermal disinfected bone human femoral head combined with platelet rich plasma, recombinant human bone morphogenetic protein 2, and zoledronic acid

Although this work reports some interesting results on biofactors-combined scaffolds for bone regeneration, major revision is required to improve the quality of this study, as commented below:

1. please describe in detail how they prepared implants, and how they combined the PRP and rhBMP.

2. authors need to mark angiogenesis and osteoclasts as well in the histological images.

3. bone regeneration area is rapidly growing, thus some of the recent approaches of bone scaffolds should be cited and discussed within text, as below:

- Pro-angiogenic and osteogenic composite scaffolds of fibrin, alginate and calcium phosphate for bone tissue engineering, 2021,

- An injectable and self-healing hydrogel with dual physical crosslinking for in-situ bone formation, 2023

- Comparison of bone formation mediated by bone morphogenetic protein delivered by nanoclay gels with clinical techniques (autograft and InductOs®) in an ovine bone model, 2022.

- Nanofibers Regulate Single Bone Marrow Stem Cell Osteogenesis via FAK/RhoA/YAP1 Pathway, 2018

4. authors need to discuss more on the effects of each factor, PRP and rhBMP, and their synergistic role if possible, when compared to other previous works also.

n/a

Author Response

Thank you for your constructive feedback and the insightful questions you raised about our manuscript. Your comments have given us the opportunity to improve our study and provide better clarity to our readers. Please find below our detailed responses to your queries:

Q1. please describe in detail how they prepared implants, and how they combined the PRP and rhBMP. 

Answer: Thank you for your comment. We incorporated the required details into the description of the outcomes.

Q2. authors need to mark angiogenesis and osteoclasts as well in the histological images.

Answer: Thank you for your comment. It has been corrected in the manuscript.

Q3. bone regeneration area is rapidly growing, thus some of the recent approaches of bone scaffolds should be cited and discussed within text, as below:

Pro-angiogenic and osteogenic composite scaffolds of fibrin, alginate and calcium phosphate for bone tissue engineering, 2021,

- An injectable and self-healing hydrogel with dual physical crosslinking for in-situ bone formation, 2023

Comparison of bone formation mediated by bone morphogenetic protein delivered by nanoclay gels with clinical techniques (autograft and InductOs®) in an ovine bone model, 2022. 

- Nanofibers Regulate Single Bone Marrow Stem Cell Osteogenesis via FAK/RhoA/YAP1 Pathway, 2018

Answer: Thank you for your comment. It has been corrected in the manuscript.

Q4. authors need to discuss more on the effects of each factor, PRP and rhBMP, and their synergistic role if possible, when compared to other previous works also.

Answer: Thank you for your comment.

Numerous studies have been conducted and documented in the literature concerning the histological facets of PRP's effectiveness when combined with various grafting materials, which have enriched our discussion. We have further expanded this knowledge base by incorporating data on the combined use of PRP and rhBMP-2 with thermally treated bone.

Thank you once again for your valuable feedback. Your contribution has helped us to improve our work and present it in a way that is more comprehensive and beneficial for our readers. We look forward to hearing any other comments or suggestions you might have, and we hope that you will find the revised manuscript acceptable for publication.

Round 2

Reviewer 2 Report

While the authors did provide a revision of their manuscript, it is important to note that the quality of their revisions was not satisfactory. They made some changes by simply replied to my concerns and suggestions without conducting any additional experiments as I had suggested, their explanation for their findings still seems farfetched and fails to address my concerns adequately. Therefore, I continue to insist on my original recommendation to reject this work.

moderate

Author Response

Dear Reviewer,

Thank you for providing further feedback on our revised manuscript. We appreciate your thorough evaluation, although we regret that our revisions did not meet your expectations. We would like to address your concerns and clarify some points.

The primary aim of our study was to present the histological characteristics of bone tissue regeneration. While we understand your recommendation to conduct additional experiments, such as micro-CT imaging, we would like to clarify that these investigations were beyond the scope of the current study. However, based on the findings of our current study, we are currently planning further research focusing on the combined application of thermally treated bone allograft with osteoinductive agents.  Specifically, we are in the process of designing experiments focused on the combined use of thermally treated bone allograft with osteoinductive agents. In these upcoming studies, we intend to include micro-CT imaging, immunohistochemical analyses using anti-Osteocalcin antibody, anti-CD31 antibody, and anti-Osteopontin antibody. These additions will provide a more comprehensive evaluation of bone healing and regeneration.

Regarding the bone allograft preparation process, we appreciate your feedback and have made significant revisions to describe the method in detail, particularly according to the Marburg system. We have also provided a comprehensive explanation of how the histological evaluation of bone defect healing was conducted.

We acknowledge the limitations of our study, especially the sole reliance on histological examination and the absence of micro-CT imaging. We have explicitly mentioned these limitations in the revised manuscript, and we recognize that these additional techniques could have further enhanced our understanding of bone regeneration.

We appreciate your insights and comments, which have helped improve the quality and clarity of our manuscript. While we understand your continued recommendation for rejection, we kindly request you to reconsider your decision, considering the revisions we have made to address your concerns and the potential value our study offers in the context of histological bone tissue regeneration.

Thank you for your time and consideration. We remain open to further discussions and addressing any additional concerns you may have.